Seasonal availability of edible underground and aboveground carbohydrate resources to human foragers on the Cape south coast, South Africa

De Vynck Jan C. 1 jandevynck@vodamail.co.za
Cowling Richard M. 1
Potts Alastair J. 1
Marean Curtis W. 1 2
1 Centre for Coastal Palaeosciences, Nelson Mandela Metropolitan University , Port Elizabeth, Eastern Cape , South Africa
2 Institute of Human Origins, School of Human Evolution and Social Change, Arizona State University , Tempe, Arizona , United States
Barrett Louise
Electronic publication date: 2016 Feb 18
Publication date: 2016
Volume: 4
Electronic Location ID: e1679
Received 2015 Sep 15; Accepted 2016 Jan 20
Copyright: © 2016 De Vynck et al.
Copyright year: 2016
Copyright holder: De Vynck et al.
License: This is an open access article distributed under the terms of the Creative Commons Attribution License, which permits unrestricted use, distribution, reproduction and adaptation in any medium and for any purpose provided that it is properly attributed. For attribution, the original author(s), title, publication source (PeerJ) and either DOI or URL of the article must be cited.
License URL: https://creativecommons.org/licenses/by/4.0/

Keywords: Cape Floristic Region, Cognitively modern humans, Edible fruit, Geophytes, Renosterveld, Strandveld, Fynbos, Underground storage organs, Hunter-gatherers

Funding: National Research Foundation 73664 Funding for this study was provided by the National Research Foundation (Grant No: 73664), Nelson Mandela Metropolitan University, and the Oppenheimer Memorial Trust. CWM received a grant from the John Templeton Foundation to the Institute of Human Origins at Arizona State University. The funders had no role in study design, data collection and analysis, decision to publish, or preparation of the manuscript.

==============================
The coastal environments of South Africa’s Cape Floristic Region (CFR) provide some of the earliest and most abundant evidence for the emergence of cognitively modern humans. In particular, the south coast of the CFR provided a uniquely diverse resource base for hunter-gatherers, which included marine shellfish, game, and carbohydrate-bearing plants, especially those with Underground Storage Organs (USOs). It has been hypothesized that these resources underpinned the continuity of human occupation in the region since the Middle Pleistocene. Very little research has been conducted on the foraging potential of carbohydrate resources in the CFR. This study focuses on the seasonal availability of plants with edible carbohydrates at six-weekly intervals over a two-year period in four vegetation types on South Africa’s Cape south coast. Different plant species were considered available to foragers if the edible carbohydrate was directly (i.e. above-ground edible portions) or indirectly (above-ground indications to below-ground edible portions) visible to an expert botanist familiar with this landscape. A total of 52 edible plant species were recorded across all vegetation types. Of these, 33 species were geophytes with edible USOs and 21 species had aboveground edible carbohydrates. Limestone Fynbos had the richest flora, followed by Strandveld, Renosterveld and lastly, Sand Fynbos. The availability of plant species differed across vegetation types and between survey years. The number of available USO species was highest for a six-month period from winter to early summer (Jul–Dec) across all vegetation types. Months of lowest species’ availability were in mid-summer to early autumn (Jan–Apr); the early winter (May–Jun) values were variable, being highest in Limestone Fynbos. However, even during the late summer carbohydrate “crunch,” 25 carbohydrate bearing species were visible across the four vegetation types. To establish a robust resource landscape will require additional spatial mapping of plant species abundances. Nonetheless, our results demonstrate that plant-based carbohydrate resources available to Stone Age foragers of the Cape south coast, especially USOs belonging to the Iridaceae family, are likely to have comprised a reliable and nutritious source of calories over most of the year.

Introduction

The Cape south coast has likely been occupied by hominins for at least the last 1 million years and the earliest archaeological remains attributable to Homo sapiens dates to approximately 160,000 years ago (Jerardino & Marean, 2010). The archaeological record on the Cape south coast for the period between 160,000 and 50,000 years is unusually data rich and well dated. This region and time period is crucial to our understanding of modern human origins as it provides some of the earliest evidence for the emergence of complex behaviours associated with cognitively modern humans, making it highly significant to human origins studies (Marean et al., 2014). The archaeological record documents human occupation during periods of glacial maxima, such as Marine Isotope Stages 6 and 4, when climatic conditions over much of the rest of Africa were too harsh for human occupation or could sustain only very small populations (Bar-Matthews et al., 2010; Marean et al., 2014). It has been hypothesised that the richness of the record and continuity of occupation along the Cape south coast, and the Cape Floristic Region (CFR) more generally, is a consequence of an unusually rich resource base unique to this area (Parkington, 2001; Parkington, 2003; Parkington, 2006; Marean, 2010; Marean, 2011). The coastline in this region offers a highly productive inter-tidal zone for shellfish collection for human foragers. During glacial phases, an extensive plain was exposed off the current coast that supported a diverse plains game fauna, which would have offered excellent hunting opportunities (Klein, 1983; Marean, 2010). The CFR is also home to a globally exceptional flora with many species offering harvestable edible carbohydrates (Deacon, 1970; Parkington & Poggenpoel, 1971; Van Wyk & Gericke, 2000; Van Wyk, 2002; Schwegler, 2003; Dominy et al., 2008; De Vynck, Van Wyk & Cowling, 2016). These include geophytic Underground Storage Organs (USOs) that are both highly diverse and locally abundant (Goldblatt, 1978; Procheş, Cowling & du Preez, 2005; Procheş et al., 2006; Singels et al., 2015), as well as many species with aboveground carbohydrates such as fruit, vegetables, seed pods and seeds (De Vynck, Van Wyk & Cowling, 2016). Together these resources may have provided a complementary set of protein and carbohydrate-rich foods to a human forager, thus explaining the continuity of human occupation through glacial maxima.

However, to date the actual availability and productivity of these potential food resources to a human forager has been largely based on conjecture. For example, Marean’s (2010) argument that CFR geophyte diversity directly resulted in a wide range of collectable plant foods would not hold if most of those plants were poisonous, very low in caloric returns, very costly to procure, or unavailable for large parts of the year. To better understand the record for hunter-gatherer foraging in this region, we must develop robust understandings of the foraging potential of the plant foods. This includes analyses of their nutritional character (Kyriacou et al., 2014), availability in the landscape and importantly their seasonal availability to a forager.

This study solely focuses on the temporal availability of edible carbohydrates from a range of plant species in four vegetation types on South Africa’s Cape south coast. Availability is estimated based on the visibility of plant parts that directly or indirectly lead to edible carbohydrates. We focus on visibility as a proxy for availability as human foragers primarily rely on sight and have a poor sense of smell. Thus, above-ground edible carbohydrates may be readily visible and available when plants are in fruit, but below-ground carbohydrates can only be found if there are above-ground indicators such as leaves, flowers, or dried stalks. To the best of our knowledge at this time, above ground visibility of plant foliage is the most reliable determinant that a human forager could positively identify and extract the underground resource. The ultimate goal is to combine these observations with studies of nutrition, abundance estimates and return rates, so as to contribute to the resourcescape for a paleoscape model for the Cape south coast (Marean et al., 2015). More generally, our paper adds to a growing literature on the importance of geophytes and aboveground carbohydrates to hunter-gatherer diet worldwide (Kaye & Moodie, 1978; Hatley & Kappelman, 1980; Vincent, 1985; Murray et al., 2001; Bird, Bliege Bird & Parker, 2005; Bliege Bird et al., 2008), and expands the range of variation of those data to a region that is megadiverse in plant species yet relatively unstudied in regards to plants, and geophytes in particular, as a food resource.

Methods

Study area

The study area is situated in the coastal plain between the Breede and Gouritz rivers on the Cape south coast (Fig. 1). The rainfall regime shows little seasonality and rain may fall at any time of the year although slight rainfall peaks are observed in March–April and with more pronounced peaks during August and October–November (Engelbrecht et al., 2014). The overall climate of the study area is semi-arid to sub-humid with annual rainfall ranging from 350 to 550 mm. The three summer months (Dec–Feb) are the most stressful for plant growth, owing to generally lower rainfall and persistently higher temperatures.

Figure 1 (A) The location and major vegetation types of the study region and the plot localities [1: Renosterveld (purple); 2: Limestone Fynbos (green); 3: Sand Fynbos (yellow); and 4: Strandveld (orange; restricted to the coastal margin; see Table S1 for further plot details)]. (B) A Walter-Leith climate diagram from the town of Still Bay (∼5 km from plot 2). (C) Photos taken within the four plots in the different vegetation types.

Vegetation of the Cape coastal lowlands is under strong edaphic control (Thwaites & Cowling, 1988; Rebelo et al., 1991; Cowling & Potts, 2015) and the study area has a wide range of geologies which generate different soil types. These include Table Mountain Group sandstones (visible on the coast), Bokkeveld shales (exposed on the inland margin of the study area), Cretaceous Enon Formation conglomerates and mudstones (∼25 km from the coast), and Bredasdorp Formation limestones (Rogers, 1984; Malan, 1987). In addition, near the coastal margin aeolian sands of marine origin mantle the geology and this varies in pH with age; younger sands are alkaline and older sands are leached and acidic (Rebelo et al., 1991; Abanda, Compton & Hannigan, 2011). Shale- and mudstone-derived soils are moderately fertile, while those associated with leached sands are infertile. The calcareous sands associated with limestone, calcrete and coastal dunes are also relatively infertile due to their high alkalinity and subsequent low levels of plant-available phosphorus (Thwaites & Cowling, 1988).

The Cape Floristic Region (CFR) is home to four biomes, namely Fynbos, Renosterveld, Forest and Subtropical Thicket (Bergh et al., 2014). The vegetation types chosen for this study are representative of dissected coastal margin of the southern CFR, namely Fynbos (two types), Renosterveld and Subtropical Thicket (Rebelo et al., 1991). Generally, Fynbos communities are richest in species, especially local endemics (Cowling, 1990); Renosterveld harbours a high diversity of USO-bearing geophytes (Procheş et al., 2006), while Subtropical Thicket has a high diversity of fruit-bearing trees and shrubs (Cowling et al., 1997).

This study monitored the phenological phase of edible plants–periodic life cycle events–growing in single plots located in Renosterveld, Sand Fynbos, Limestone Fynbos, and Strandveld (a form of Subtropical Thicket). Renosterveld occurs on the relatively fertile and fine-grained soils derived from shales and mudstones, and is a fire-prone grassy shrubland often dominated by Elytropappus rhinocerotis (renosterbos). Sand Fynbos occurs on infertile acid soils and is a fire-prone heath-like shrubland, characterised by the presence of Restionaceae and Proteaceae. Limestone-derived soils support Limestone Fynbos, a highly endemic-rich vegetation type (Willis et al., 1996). Marine sands are associated with Subtropical thicket, either in its solid form or as thicket clumps in a matrix of Fynbos. This vegetation is colloquially known as Strandveld. Plant compositional change, or beta diversity, between these edaphically differentiated vegetation types is extremely high; consequently few species are shared among these four vegetation types and regional-scale plant richness is very high (Cowling, 1990).

Data collection

The monitoring period was from May 2010 until April 2012. In the year prior to monitoring (2009), the rainfall was far below average across all plots (∼70% of the mean annual rainfall). The effects of the previously dry year were still evident when monitoring started. Above average rainfall was experienced over the two years of monitoring (see Fig. S1 in Supplementary Materials for climate diagrams and Fig. S2 for spatial relation of weather stations to survey plots).

Monitoring plots were located in representative areas of each of the major vegetation types (one plot per vegetation type) described above: Renosterveld, Sand Fynbos, Limestone Fynbos, and Strandveld (Fig. 1). These plots were located within protected areas and were considered to be in a pristine condition. Further biophysical data for the plots are provided in the Supplementary Materials (Table S1). The Sand Fynbos site had burnt four years before the start of the survey period and this would likely have enhanced the visibility of USO species (Deacon, 1993), many of which flower more profusely in the early post-fire years (Le Maitre & Midgley, 1992).

Each plot was divided into six 20 × 300 m transects (3.6 ha in total). Monitoring consisted of surveying each plot every six weeks over a two-year period by one of the authors (JC De Vynck), who is a trained field botanist familiar with the vegetation in this landscape. Along each transect, the following were counted: 1) individuals of species bearing Underground Storage Organs (USOs) which would be apparent to a forager (i.e. in a phenological phase where one or more aboveground organs were visible) and, 2) individuals of species with edible aboveground carbohydrates; these included fruits, leaves, seed pods, seeds, and inflorescences. In our sampling approach we adopted a forager’s perspective by only including species known to be edible (De Vynck, Van Wyk & Cowling, 2016) and excluding any plants considered too small to harvest. We included as edible any USOs that required cooking in order to render them edible (e.g. tubers of Rhoicissus digitata and corms of Chasmanthe aethiopica and Watsonia spp.) (Wells, 1965; Parkington & Poggenpoel, 1971; Deacon, 1976; Deacon, 1979; Liengme, 1987; Opperman & Heydenrych, 1990; Skead, Manning & Anthony, 2009).

As stated above, availability of edible carbohydrates was based on direct or indirect visibility of the resource. For example, USOs in their dormant phase are not visible aboveground and can therefore not be procured by foragers. However, visible phases of USOs include any above indicator of their presence, such as green leaves, flowers or dry wilted leaves. The nutritional content may vary through these phases, but is not the focus of this study. The majority of species with aboveground carbohydrates (e.g. fruiting species) are perennially visible, but were only recorded as ‘available’ to foragers when edible carbohydrates were visible.

Data analysis

For each plot, the number of visible, and hence available, species with edible carbohydrates over time was determined; this was calculated as the number of individuals observed in a given survey that would provide access to edible carbohydrates divided by the maximum number of individuals observed across all surveys (for a given species within a plot). In order to calculate the number of species with edible carbohydrates for a given period within a plot, the continuous proportions of individuals with edible carbohydrates per species were converted to binary presence or absence categories using a 10% threshold. Thus, we considered edible carbohydrates offered by a given species readily visible and available in the landscape if at least 10% of the maximum observed individuals were visible with direct or indirect access to carbohydrates. The number of species with edible carbohydrate considered available in each plot through time was calculated. All analyses were conducted in R version 2.15 (R Development Core Team, 2014).

Results

Within the four 3.6 ha plots spread across the four vegetation types, 52 edible plant species were recorded. Of these, 33 species were geophytes with edible Underground Storage Organs (USOs) and 21 species had aboveground edible carbohydrates (Table 1; see Tables S3 and S4 in Supplementary Materials for full species list per type). Note that some species had more than one edible part. Richness of edible species varied across the vegetation types (Table 1): Limestone Fynbos had the richest flora, followed by Strandveld, Renosterveld and lastly, Sand Fynbos.

Table 1 Summary of edible species in 3.6 ha plots situated in four dominant vegetation types along the Cape south coast.

	USOs	Fruit	Other1	All	
Renosterveld	8	6	2	16	
Sand Fynbos	5	4	1	10	
Limestone Fynbos	21	11	7	39	
Strandveld	15	8	5	28	
Across all types2	33	14	8	52	
Notes:

1 ‘Other’ includes species with edible: seed pods, seeds, leaves, and inflorescences.

2 Note that this is the number of unique species (i.e. some species are shared between vegetation types or have more than one edible part).

Species varied in the length of time they were available through the year (Fig. 2; see Supplementary Materials for full list of species phenology diagrams (Table S2.1 to S2.8) and phenphase synchronicity among the species (Fig. S3)). Species with USOs were available for longer periods of the year relative to those with edible aboveground carbohydrates. The availability of USO species differed across vegetation types and between survey years (Fig. 3). Nonetheless, the number of available USO species was highest for a six-month period from winter to early summer (Jul–Dec) across all vegetation types. Months of lowest species’ availability were in mid-summer to early autumn (Jan–Apr); the early winter (May–Jun) values were variable, being highest in Limestone Fynbos. In the wetter second year, the summer “crunch” period–where few USO species were available–was at least one month shorter than in the first year. The number of species with available edible aboveground carbohydrates also varied across vegetation types and sample years. Species richness peaked in spring (Sep–Nov) for all vegetation types; relatively high availability extended into summer (Dec–Feb) but autumn and early winter were lean months for harvesting aboveground carbohydrates in all vegetation types, especially Renosterveld and Sand Fynbos. The presence of two Carpobrotus species, which bear ripe fruits during the drier months, was a key factor for the extension of aboveground availability period in Limestone Fynbos.

Figure 2 A breakdown of the number of months in which different plant species (with edible carbohydrates) are visible through the year separated into (A) Underground Storage Organs (USOs) and (B) aboveground carbohydrates (e.g fruit, seed pods, seeds, leaves or inflorescences).

Figure 3 The seasonal availability of edible species visible to a human forager in four vegetation types dominant along the Cape south coast.

Underground storage organs are geophytes that have tubers, corms, bulbs or rhizomes, while above-ground carbohydrates includes species with edible fruit, seed pods, seeds, leaves or inflorescences. The number of new species observed since the previous survey is shown in each circle; this provides an indication of species turnover.

An impressive 25 species provided edible carbohydtate during the late summer (Feb–Mar) “crunch period (Table 2). Twelve of these were USOs and Limestone Fynbos supported the most species (16) with available carbohydrates present at this time.

Table 2 Species available during the ‘carbohydrate-crunch’ late summer period (February–March) in both survey years.

Vegetation type	Carbohydrate category	Species	
Renosterveld	Underground storage organ	Babiana patula; Cyphia digitata; Watsonia meriana	
	Aboveground	Diospyros dichrophylla (fruit); Osyris compressa (fruit); Sideroxylon inerme (fruit)	
Limestone Fynbos	Underground storage organ	Cyphia digitata; Gladiolus exilis; Pelargonium lobatum; Pelargonium triste; Rhoicissus digitata; Watsonia fergusoniae	
	Aboveground	Carissa bispinosa (fruit); Carpobrotus accinaciformis (fruit); Carpobrotus edulis (fruit); Cynanchum obtusifolium (seedpods); Euclea racemosa (fruit); Osyris compressa (fruit); Searsia glauca (fruit); Sideroxylon inerme (fruit); Tetragonia decumbens (leaves); Zygophyllum morgsana (seed)	
Sand Fynbos	Underground storage organ	Gladiolus guthriei; Watsonia fourcadei	
	Aboveground	Carpobrotus edulis	
Strandveld	Underground storage organ	Chasmanhte aethiopica; Ferraria crispa; Rhoicissus digitata	
	Aboveground	Carissa bispinosa (fruit); Carpobrotus accinaciformis (fruit); Osteospermum moniliferum (fruit); Schotia afra (seed); Tetragonia decumbens (leaves)	

Discussion

Substantial archaeological evidence exists for the use of Underground Storage Organs (USOs), fruits and leaves by Late Stone Age peoples in southern Africa (Deacon & Deacon, 1963; Parkington & Poggenpoel, 1971; Deacon, 1970; Deacon, 1976; Deacon, 1984; Opperman & Heydenrych, 1990; Deacon & Deacon, 1999). This evidence is substantiated by direct observations of contemporary hunter-gatherer communities in Africa (Lee, 1969; Lee, 1973; Lee, 1984; Silberbauer, 1981; Youngblood, 2004; Berbesque & Marlowe, 2009; Marlowe & Berbesque, 2009). The diversity and abundance of edible plants, especially USOs, along the Cape coast, together with a rich source of both marine and terrestrial based protein, has been hypothesised to be key components facilitating the persistence of Middle Stone Age (MSA) people in the region during glacial phases when other African regions may have been resource poor (Marean, 2010). However, very little research has been conducted on the potential availability of food plants to hunter-gatherers on the Cape south coast to corroborate this hypothesis. In the same study area, Singels et al. (2015) found surprisingly high edible biomass values for USOs (maximum values range from 600 kg/ha in Sand Fynbos to 5,000 kg/ha in Limestone Fynbos), although these were restricted to occasional biomass hotspots within a matrix of much lower biomass. Also, these USO hotspots were found within all vegetation types. Here we address the temporal availability of belowground (i.e. USOs) and aboveground sources of carbohydrates across the four principal vegetation types of the Cape south coast. We use this to speculate on the importance of carbohydrates as fallback foods for coastal hunter-gatherers, and what role this may have played in the emergence of cognitively modern people in the region (Marean, 2010).

The number of species with edible carbohydrate resources that are visible and available to foragers was highest between winter and early summer in the study area. This is consistent with the dominant cool-season phenology of plants in the Cape Floristic Region (Pierce, 1984). This six-month period provides a diversity of USOs associated with corms belonging to petaloid geophytes, mostly members of the Iridaceae (e.g. Babiana, Freesia, Gladiolus, Watsonia). These species provide relatively large (10–100 g) starch-rich and low-fibre food parcels that are inexpensive to harvest (Parkington, 1977; Deacon, 1993; Singels et al., 2015), and many do not require cooking for digestion (Youngblood, 2004; Dominy et al., 2008; J. De Vynck, personal observations, 2011). Also available during the cooler and mostly wetter months are fruits borne largely by Subtropical thicket species (e.g. Carissa, Diospyros, Olea, Searsia) as well as leaf crops (Trachyandra spp.). There are currently no data on the biomass, nutritional value and foraging returns for aboveground sources of carbohydrates in the Cape Floristic Region. Fruit loads of mature thicket shrubs and trees range from tens of thousands of fruits per plant for Sideroxylon inerme (fruit diameter 10 mm) and Searsia spp (3 mm) to fewer than 100 fruits for Euclea racemosa (7 mm), Cassine tetragona (8 mm) and Osyris compressa (20 mm) (Cowling et al., 1997). Mat-forming Carpobrotus species may bear several tens of large (35 mm diameter) fruits (J. De Vynck, personal observations, 2011).

Late summer to early autumn periods have considerably fewer available edible species than in the other times of the year. This is a period when all traces of leaves and inflorescences of the dominant deciduous geophyte component have disappeared (Deacon, 1993). However, even during this relatively warm and dry period, we recorded some 25 available species across the four vegetation types (Table 2). These include USOs such as hysteranthous, autumn-flowering Gladiolus (cormous) and Pelargonium (tuberous) spp, the corms of evergreen Watsonia spp., and the fibrous tubers of the evergreen liana, Rhoicissus digitata. Also apparent are the fruits Carpobrotus spp, the fruits of many thicket shrubs and trees, and the leaf crop, Tetragonia decumbens. Nonetheless, the late summer–early autumn months could represent a carbohydrate “crunch” for foragers: at this time the number of edible plant species is at its lowest and the high-biomass items available to foragers (e.g. Pelargonium spp., R. digitata) are fibrous and require cooking for digestion (Deacon, 1995; Wandsnider, 1997; Laden & Wrangham, 2005; Dominy et al., 2008; Schnorr et al., 2015).

Overall, the plant-based carbohydrate resources available to Stone Age foragers of the Cape south coast, especially USOs belonging to the Iridaceae (Deacon, 1976; Deacon, 1993), are likely to have comprised a reliable and nutritious source of calories over most of the year. Moreover, availability of USOs showed little between-year variation, most likely due the existence of sufficient storage reserves to enable at least leaf growth every year (Ruiters & McKenzie, 1994) despite variation in rainfall. In an assessment of foraging potential of six USO species growing in our study area, Singels et al. (2015) showed that 50% of foraging events conducted yielded enough calories to meet the daily requirements of a hunter-gatherer of small stature within two hours.

The juxtaposition within a 10 km foraging radius of four major vegetation types, belonging to three regional biomes (Fynbos, Renosterveld and Subtropical Thicket; (Bergh et al., 2014)), would have enabled humans to forage in very different resourcescapes on a daily basis. While the Limestone Fynbos and Strandveld–the two vegetation types closest to the coast–are likely to have offered the best foraging returns for much of the year, Renosterveld provides an abundance of Iridaceae corms in the spring and Sand Fynbos harbours evergreen Watsonia spp., which can be harvested during the late summer-autumn “crunch” (Singels et al., 2015). Ethnographic evidence suggest that the harvesting of Iridaceae corms (uintjies) in spring was an important event for the San of the Cape west coast (Van Vuuren, 2014).

Given the temporal and spatial availability of edible plant species in the Cape, we argue that is highly likely that USOs, fruit, seedpods, seeds, inflorescences and leaf crops were harvested as fallback foods by Stone Age people living in this region. The likely preferred food for south Cape coastal hunter-gatherers comprised the region’s diverse and abundant marine resources (Marean et al., 2007; Jerardino & Marean, 2010; Parkington, 2010), and a diverse ungulate plains fauna, including in the Pleistocene, several species of now extinct megafauna, associated with the submerged Agulhas Bank (Klein, 1983; Parkington, 2001; Parkington, 2003; Matthews, Marean & Nilssen, 2009; Marean, 2010; Faith, 2011). However, these resources were not always available to harvesters and hunters, and the contraction and expansion of the Agulhas Plain ecosystem and its ungulate communities must have been a major driver of changing foraging patterns on the south coast (Marean et al., 2014). It has been hypothesized that the mammal fauna formed a migratory community that moved west during the winter rains and east to intercept the summer rains. Thus, the local abundance of many of the larger ungulates may have plummeted during the winter months when populations migrated west to graze winter-growing grasses of the west coast. Marine invertebrates, harvested from the intertidal, comprised the most reliable and accessible source of protein for hunter-gatherers living on the Cape south coast (Marean, 2011). Evidence for their use has been found in MSA sites such as Pinnacle Point (PP) 13B dating back to ∼160 ka (Marean et al., 2007; Jerardino & Marean, 2010) and at early modern human sites that date between 110–150 ka such as Blombos Cave (Henshilwood et al., 2001; Langejans et al., 2012), and Klasies River Mouth (Voigt, 1973; Thackeray, 1988). Late Stone Age sites suggest an increase in the intensity of intertidal foraging (Marean et al., 2014) and indications of resource depletion (Klein & Steele, 2013). Using experienced foragers of Khoe-San descent, (De Vynck et al., in press) showed exceptionally high peak return rates (∼4,500 kcal hr−1) from the Cape south coast intertidal under ideal harvesting conditions. However, owing to tidal constraints, and the fierce sea conditions experienced there, harvesting was only possible for 10 days a month, for 2–3 hours on each day; lowest returns were recorded in winter and spring–a time of strong winds and high seas–and highest returns in summer and autumn, when sea conditions were calmer (De Vynck et al., in press). Consequently, there would have been periods of various lengths–ranging from days to weeks–when hunter-gatherers depended on, or fell back upon carbohydrates for sustenance. As pointed out above, the winter and early spring months likely coincided with a scarcity of protein but an abundance of carbohydrates. At these times, plant carbohydrates, especially USOs, may have comprised 100% of dietary intake, which would categorise them as a staple fallback food (Marshall & Wrangham, 2007).

It has been hypothesized that the persistence of a small group of hominins on the Cape south coast–as opposed to their widespread extinction elsewhere in Africa during Marine Isotope Stage 6 (MIS6, 193,000–125,000 BP) (Foley, 1998; Lahr & Foley, 1998; Fagundes et al., 2007; Basell, 2008; Masson-Delmotte et al., 2010)–was a consequence of the Cape’s relatively moderate climate during the largely glacial MIS6 and its rich and diverse resource base. The persistently warm Agulhas Current reduced the regional impact of glacial cooling substantially (Negre et al., 2010; Zahn et al., 2010). Marean (2010) has hypothesised that during strong glacial environments, such as those experienced in MIS6, the Cape south coast provided a unique juxtaposition of resources important for hominin persistence, namely a diverse USO flora and a rich and productive marine ecosystem. At that time the exposed Agulhas Plain (Fisher, Bar-Matthews & Marean, 2010) was mantled in substrata that likely supported Renosterveld, Limestone Fynbos and Strandveld (Cawthra et al., 2015), offering a wide array of USOs, fruit and leaf crops which would comprise reliable fallback foods when it was not possible to forage in the intertidal and game was scarce. The cognitive challenges of exploiting marine resources (e.g. comprehending lunar cycles), and defending them against competition from adjacent groups, led to a coastal adaptation that may have contributed to the emergence of cognitively modern Homo sapiens (Marean, 2011). Similarly, the ability to recognise which and when vegetation types are most productive for carbohydrates, identifying hotspots of productivity and distinguishing between edible and toxic USOs, must have been challenging (Deacon, 1995).

Here we have established the temporal availability of plant species with edible carbohydrates across four dominant vegetation types along the south coast. Much additional research must be done to evaluate more comprehensively the role of above- and belowground carbohydrates in the ecology and evolution of the human lineage in the Cape Floristic Region and elsewhere. Work is currently underway to establish the return rates of carbohydrate resources harvested by contemporary subjects of Khoe-San descent, in the different vegetation types and in different seasons; and on the rates of depletion of resources in successively harvested areas. This needs to be complemented with data on the nutritional value of the consumed parts of the species selected. Ultimately, we aim to use these data to populate the carbohydrate resourcescape in an agent-based model aimed at predicting the effects of spatial and temporal variability–governed by changes in climate and the resource base over the seasonal cycle as well as the glacial-interglacial cycle of the Pleistocene–on the population size and structure, mobility, social organization, territoriality, and technology of Cape hunter-gatherers (Marean et al., 2015).

Supplemental Information

Supplemental Information 1 Meta data.

Meta data for raw data file.

Click here for additional data file.

Supplemental Information 2 Raw data collected six-weekly over two years in the four primary vegetation types, southern Cape, South Africa.

The raw data are six-weekly counts, over two years, of edible plants with Underground Storage Organs and aboveground edibles. These surveys were performed in predesignated 3.6 hectare plots of the four primary vegetation types of the southern Cape, South Africa.

Click here for additional data file.

Supplemental Information 3 Supplementary Material Methods.

An index of all supplementary tables and figures. Phenological phase synchronicity is explained in detail and the methods which were employed in creating Fig. S3. References for the phenological phase synchronicity description.

Click here for additional data file.

Supplemental Information 4 Biophysical information.

Biophysical information for the four study sites.

Click here for additional data file.

Supplemental Information 5 Phenology diagrams.

Phenology diagrams for plant species with Underground Storage Organs (USOs) and for fruiting plant species (aboveground carbohydrate resources) of the four primary vegetation types of the south Cape lowlands to coastal margin.

Click here for additional data file.

Supplemental Information 6 Total species list.

Total species list of USOs and fruiting species (aboveground carbohydrate resources) and their acronyms (used in Fig. S3) encountered in the phenology survey of plots within four vegetation types of the southern Cape lowlands to coastal margin.

Click here for additional data file.

Supplemental Information 7 Total species list per vegetation type.

Species list of USOs and fruiting species (aboveground carbohydrate resources), and their acronyms, encountered in the phenology survey list of plots within four dominant vegetation types in the southern Cape lowlands to coastal margin. Acronyms relate to those used in Fig. S3.

Click here for additional data file.

Supplemental Information 8 Climate diagrams.

Climate diagrams showing temperature and rainfall patterns for the study sites during the survey period (May 2010–April 2012). Temperature and rainfall axes and shading follow Walter-Lieth conventions. Mean values of temperature and rainfall for the period are shown in parentheses. The positions of weather stations relative to survey plots are shown in Fig. S2.

Click here for additional data file.

Supplemental Information 9 Climate map.

Locations of the vegetation survey plots and the weather stations (shown in Fig. S1).

Click here for additional data file.

Supplemental Information 10 Phenological phase synchronicity.

A hierarchical classification to establish phenological phase synchronicity (specifically availability of edible carbohydrates) among plant species from four different vegetation types along the Cape south coast (South Africa). The plant species, its carbohydrate type (Underground Storage Organ [USO] or aboveground carbohydrate [i.e. fruit, vegetables, seed pods and seeds) and vegetation type are shown. The phenological phase plots show the relative proportion of individuals per species with visible edible carbohydrate through the sampling period within each cluster. The proportion of species within each phenological phase cluster per carbohydrate type (i.e. USO or aboveground) and vegetation type are also shown. Species abbreviations are Tables S3 and S4. See Supplementary Methods for full explanation of the methods used to generate this figure.

Click here for additional data file.

We thank the Cape Nature team–Rhett Heismann, Jean Du Plessis and Leandi Wessels–for access, support, information and GIS assistance. We also thank the Hessequa Municipality, and in particular Hendrik Visser, for their help and support. The authors are also grateful for the climate data supplied by South African Weather Service. Peter Henzi and an anonymous reviewer provided valuable comments that improved the manuscript.

Additional Information and Declarations

Competing Interests

Author Contributions

Data Deposition

Richard M. Cowling is an Academic Editor for PeerJ. The opinions expressed in this publication are those of the author(s) and do not necessarily reflect the views of the John Templeton Foundation.

Jan C. De Vynck conceived and designed the experiments, performed the experiments, analyzed the data, contributed reagents/materials/analysis tools, wrote the paper, prepared figures and/or tables, reviewed drafts of the paper.

Richard M. Cowling conceived and designed the experiments, analyzed the data, contributed reagents/materials/analysis tools, wrote the paper, prepared figures and/or tables, reviewed drafts of the paper.

Alastair J. Potts analyzed the data, wrote the paper, prepared figures and/or tables, reviewed drafts of the paper.

Curtis W. Marean wrote the paper, reviewed drafts of the paper.

The following information was supplied regarding data availability:

The raw data was supplied as Supplemental Dataset files.

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
