# Peer review of "Seasonal availability of edible underground and aboveground carbohydrate resources to human foragers on the Cape south coast, South Africa"

_PeerJ, doi:10.7717/peerj.1679_

## Round 0.1 · original submission · Major Revisions

· Academic Editor

Major Revisions

Dear Dr De Vynck,

Thank you for your submission. Your paper has now been seen by two reviewers, and I'm pleased to inform you that I'm happy to accept your paper for publication if you can revise your paper in line with their suggestions. As you will see Reviewer 2's comments are the most substantive, but these mainly ask for more detail and justification regarding your methods, and the implications of these for how you interpret your results.

·

Basic reporting

The paper reports its findings clearly and I have only a few specific comments concerning text.

l 72. It might be useful to know why this period is crucial to our understanding of human origins.

l 74. Some references about climatic conditions and population size in Africa would be helpful.

l 100. Although useful later in the discussion, Singels et al's work, especially as unpublished, is irrelevant here.

l 101-102. End sentence at 'to a forager'.

l 142. A short explanation of fynbos for a non-South African reader might be helpful.

l 216. 'the', not 'their'.

l 731-732. I suggest, 'The number of new species observed since the previous survey...'

Experimental design

This is, in essence, a descriptive study that sets out to assess the hypothesis that the spatio-temporal distribution of edible foods is consistent enough to have allowed year-round exploitation by modern hunter-gatherers. The authors structure their data to allow an additional assessment of the importance of USOs in this exploitation. The design is standard for studies based on monitoring availability and study plots are adequately sized and distributed. Analysis is straight-forward although the authors might include more explanation of their clustering technique as applied to phenophase identification (l 194), since the reference provided points only to the general technique.

Validity of the findings

The data are appropriately interpreted and, with one exception, speculation is constrained. The exception concerns the standard, tempting inference that the results necessarily point to sophisticated cognition. Here (l 61; l 377-380), the argument concerns the assumed cognitive demands of recognising hotspots and identifying edible plants. Given that baboons move efficiently between habitat types and feed on most of the same foods in that part of the Cape (unpubl. data), including shellfish (Hall 1965), and rely almost entirely on extractive foraging of USOs in the Drakensberg grasslands, where edible sedge corms need to be distinguished by their dried leaves from 15-20 non-edible spp. in the immediate vicinity, this claim does seem excessive. I would recommend excising it.

Reviewer 2 ·

Basic reporting

The figures could use some more work to be clearer in what they represent and the figure legends should be more descriptive so that they stand alone from the text.

Figure 3 could be enlarged, and perhaps rearranged so that the plots fall along one column instead of two so that the very important and descriptive information it contains is more easily read.

Figure 4 needs to be much better clarified in both the legend and graphical representation. Make a header on top after the dendrogram branches to delineate what the different column notations represent: i.e. "species abbreviations" (does each three letter abbreviation refer to one species or is it the full 6 letter abbreviation?); "carbohydrate location"; "vegetation type"; and "availability density plot". I don't think a reader should have to dig out the supplementary to understand a main figure, so it would also be nice to have the abbreviations listed along with the figure, or just rework the figure so that you don't need abbreviations and instead write the full species name in normal scientific nomenclature (i.e. B. patula, etc.). I realize this could make the figure too cluttered, but above all else, the authors need to do a better job of making this figure and its information stand alone without a bunch of "accessory" tables and legends and back and forth referencing in the text. In fact, the dendrogram is highly irrelevant and could be put in supplementary, while the phase density plots could be emphasized much more. I think this is also data that can and should answer "at what time of year should foragers target what vegetation types?" given the availability of different types of USO and aboveground carbohydrates. This leads me to also point out that the Fig. 4 legend needs to be more informative, and the corresponding manuscript text is similarly sparse, making this already complex figure even more enigmatic. Explain where the phenophases are represented on the figure (are they the boxed letters? I have no idea by looking at the figure alone, can you color the nodes or make them stand out more?). Also, you should bracket the density plot (I think? It is not named or referred to anywhere I can find, so I don't know what else to call this) within its correspondent dendrogram branches, or color code the tree branches and density plot together... something to make this more intuitive and easier to read.

Figure S2: Again this is an extremely informative graphic but is not well documented or intuitively illustrated, and it provides information that would be relevant for a main figure, or at least descriptive paragraph in the main text where this information is otherwise lacking. Please improve the legend at the very least. What is the difference in the rows? Why is a species named on the top row, and a family named on the bottom? Why do some have 4 rows, but no identifier? The lack of clear reporting here strikes me as somewhat hasty and incoherent.

I find Fig. 2 somewhat uninformative for the space it takes and think this is space that could be more effectively used to convey more information, such as the availability of different types of carbohydrates across the year. The authors have cited Marlowe and Berbesque 2009, and I think their figures 2 and 3 could be similarly implemented in this study, and give a much better overview of food availability per region across time.

Experimental design

In their paper titled, "Seasonal availability of edible underground and aboveground carbohydrate resources to human foragers...", the authors aim to assess the year-round availability of plant carbohydrate in 4 different vegetation types near the coast of South Africa. This was accomplished by manually surveying plots of land every six weeks for two years and tallying all edible plants that were visible along a transect. A total of 52 different species were recorded for all territories. The study design and execution is mostly sound, but for one methodological decision that is not adequately addressed, and yet has potentially profound effects on the results as reported by the authors. Specifically, the authors report that species were counted as "available to foragers" if their above ground parts were visible, and in the case of above-ground carbohydrates, viable. However, there is no criteria given to explain what constitutes "visible" and "viable". Instead, the reader must piece this all together through a bunch of disparate information, flipping through an in-press paper, supplementary figures, and sparse raw data descriptions. This is the core element of the methodology and it is essentially glossed over in a quick 2-lines and it is entirely ambiguous as to what "visible" actually means, and how it relates to plant viability, particularly for USOs. The results cannot be accepted as reported without these methods explained better. Several questions are immediately raised when I read this and should be answered in the methods:
1. What does "visible" mean?
2. Does "not visible" mean all parts of the plant are no longer showing in any way, or does this just refer to greenery, leaves and/or blooms? What about stems or vines, were they visible?
3. Is "visibility" informed by ethnographic information for how extant foragers recognize and target edible plants? I can't help but think not, because many ethnographies describe foragers as being able to recognize USO-bearing plants through associated foliage such as trees that the vines wrap around and etc.. The in-press paper has also not clarified these methods, and even makes the point that their informants have seemingly lost a lot of indigenous botanical knowledge, making it almost a point of misinformation. Given how critical this element is to the study, I am surprised that the authors have made no attempt to justify, compare, or explain their methods within the context of the greater ethnographic work on sub-Saharan foragers.
4. How well trained were the surveyors in identifying signal phenophases of viable plants? Was there any kind of practice or oversight or expert informant?
5. How do the author's survey methods of plant availability relate to previous works that attempted similar plant biomass calculations? Are they comparable? Maybe not since this study used binary classification of availability rather than mass or calorie-based calculations, which are more informative from a nutritional availability perspective.

Methods about the hierarchical clustering are vague, especially where on lines 201-203 the authors state that "Defining clusters was not performed using a strict dissimilarity threshold, but rather involved intuitive exploration of the phenograms of different potential clusters while endeavoring to maintain cluster thresholds that were fairly similar". I understand the data are not even since differences occur in number and overlap of species per site, but if "intuitive" data exploration has been conducted, then it needs to be more rigorously explained or demonstrated, otherwise it sounds like some clusters could be rather arbitrarily inclusive or exclusive.

Validity of the findings

The validity of the findings requires more attention to methodologies as described above, and further discussion or reporting about how visible "availability" relates to actual nutritional availability. As in, do low visible richness periods actually map on to low nutritionally rich periods? Can low visible richness still coincide with high nutritional availability (calorically) if the few "visible" species are highly nutritious. Therefore, the landscape could have differential "evenness" in the nutritional profile than that conveyed by "visible" profile. This is assuming that the visibility criteria used by the authors is a fair depiction of forager ability to identify available and viable species, which I question.

Additional comments

Italicize scientific nomenclature in text and tables

There is a lot of scattered methodological information that needs to be consolidated and concisely presented in the main body of the text regarding criteria for selecting "visible" plants as "present" at a given time in a given site. The authors need to spend a lot more time making this absolutely clear and preferably well supported and justified with citations.

Table 2 would be more informative with some indication of the different species’ nutritional value (even qualitative, as I realize the authors state that nutritional analyses have not been done for many above-ground species but many are fruits and seeds that are usually nutrient rich).

Given that the plots sampled are upwards of 50km away from each other, it would be helpful if you could comment on how likely it is that a foraging population situated on the coast could have access to one or more productive zone at a time from a single home base. What kinds of distances would they have to travel? Or, how often would they need to move camp to have access to a productive zone? Some comment on this would be helpful in understanding ease of access to carbohydrate resources.

Line 233, "...with carbohydrate on offer at this time" awkward and unclear phrasing.

Discussion has missed many relevant ethnographic texts pertaining to plant foraging in sub-saharan Africa, such as Silberbauer 1981 "Hunter and Habitat in the Central Kalahari" and Youngblood 2004 Economica Botany vol. 58

Dominy et al. 2008 is overused to support concepts about digestibility of foods which it does not deal with (ex: lines 282, 304), and instead Dominy et al. 2008 is about mechanical properties of African USOs and minor exploration in changes as a result of flash cooking. There are many more topical publications on both cooking and digestibility of plant foods that should be used in the relevant discussion sections. See Wandsnider 1997 JAA vol 16, Stahl 1984 Current Anth vol 25, Youngblood 2004, Wollstonecroft et al. 2008 Veget Hist Archaeobot, Laden and Wrangham 2005 JHE vol 49, Schnorr et al. 2015 AJPA for just a few examples.

---

## Round 0.2 · Minor Revisions

· Academic Editor

Minor Revisions

Thanks for your revision, which has been seen by both our previous reviewers. As you will see, there are just a few more changes suggested by our one reviewer, and I would be grateful if you would consider these and revise your paper accordingly. Once that's done, I would be delighted to accept your paper for publication. I look forward to seeing a new revision soon.
with best wishes,
Louise

·

Basic reporting

I have read the revised version and, while I was generally happy with the initial submission, I consider this revision to be improved and ready for acceptance.

Experimental design

Fine (see initial review)

Validity of the findings

Fine (see initial review)

Additional comments

Nil

Reviewer 2 ·

Basic reporting

I do not find any supplemental figures or tables cited in the main article text, only one reference to see 'supplementary materials' at line 236/237. This would be helpful as the supplemental items are very informative for this paper.

Regarding Figure 2, if this is aggregated information into an average, then that should be stated somewhere in the figure or legend, “Average number of species”.
I still think this figure would be much better represented with a count of average number of available species per month within the USO and fruit category respectively, such that the x-axis is months of the year rather than number of months.

Regarding my comment that inspiration might be taken from the presentation of data in Marlowe and Berbesque 2009, the idea would be to make a frequency plot of the metric you do have, which is presence/absence. Not necessary of course, only a suggestion to help clarify and consolidate your data presentation.

The phenodiagrams (Table S2 I believe) are still insufficiently described so as to allow a reader to understand all elements of the diagram. First, "phenological phase" (or phenophase) is not well defined in the main text (found on line 54, but undefined), and so its continued reference is somewhat unclear. Table S2 legend is therefore extremely unclear. It states that the phenophases (the colors) are shown on their own lines, but this is not true, since some lines show multiple colors, so I have to ask what do the rows mean? It is also not clear why there should be empty white boxes. Sometimes you have used black to indicate "no visibility" for one particular phenophase while others are visible, and then most of the time you have used a white box to indicate this. Please stay consistent and simple labels would help immensely.

Experimental design

Regarding my initial and main critique of the descriptive aspect of the field methods, I am satisfied that the authors have adequately explained their methodology and this is clear and acceptable for interpreting the results. However, I do think it is a bit of an over assumption to say that if primary vegetation is invisible, then human foragers would be unable to find USOs. Therefore, I think one more brief statement is necessary. Something to the effect of “To the best of our knowledge at this time, above ground visibility of the primary plant foliage is the most reliable determinant that a human forager could positively identify and extract the underground resource” This would help couch these findings not just in terms of accuracy, but also to give you the opening to make revisions or additions to your findings and methods in the future should you continue in this vein of research, which it seems likely you will. Also, perhaps you should also make some specific mention about whether visibility included associated foliage (such as other trees or shrubs that might signal existence of the hidden USO bearing plant species), or only the primary producing plant foliage.

Regarding the need for clarification on the clustering, fair point that multivariate data analysis is often intuitive, but both I and the other reviewer commented that there was too little description in the methods about how the clusters were derived, and so while this may have been an intuitive process, there must be justification or explanation of what drove that intuition. Furthermore, clustering can be validated to some degree of confidence by bootstrapping to obtain a value to support cluster inclusion/exclusion thresholds, obviating much of the subjectivity of the process.

Validity of the findings

The findings are valid and interesting and speculative statements were removed or amended.

Additional comments

I look forward to reading the results of your future efforts in mapping the resource nutritional landscape of the Cape floristic region.

---

## Round 0.3 · accepted · Accept

· Academic Editor

Accept

Thanks very much indeed for your speedy revision. I'm now delighted to accept your paper for publication.

All the very best,

Lou